# Identification of locally activated spindle-associated proteins in oocytes uncovers a phosphatase-driven mechanism

Xiang Wan, Gera Pavlova, C. Fiona Cullen, Igor Dasuzhau, Aleksandra Ciszek and Hiroyuki Ohkura*

## ABSTRACT

The meiotic spindle forms only around the chromosomes in oocytes, despite the exceptionally large volume of the cytoplasm. This spatial restriction is likely to be governed by local activation of key microtubule regulators around the chromosomes in oocytes, but the identities of these microtubule regulators and the mechanisms remain unclear. To address this, we developed a novel assay to visualise spatial regulation of spindle-associated proteins in *Drosophila* oocytes by inducing ectopic microtubule clusters. This assay identified several proteins including the TPX2 homologue Mei-38, which localised more strongly to microtubules near the chromosomes than away from them. In Mei-38, we identified a microtubule-binding domain containing a region that was also highly conserved in humans. The domain itself is regulated spatially, and contains a conserved serine and a nearby PP2A-B56-docking motif. A non-phosphorylatable mutation of this serine residue allowed the domain to localise to ectopic microtubules as well as spindle microtubules, whereas mutations in the PP2A-B56-docking motif greatly reduced the spindle localisation. As this phosphatase is concentrated at the kinetochores, it might act as a novel chromosomal signal spatially regulating spindle proteins within oocytes.

KEY WORDS: Meiosis, Spindle, Oocyte, Phosphatase, *Drosophila*

## INTRODUCTION

Owing to the absence of centrosomes, the spindle is self-assembled in the oocytes of most animal species. As the oocytes have a large volume, it is crucial to assemble one bipolar spindle around the chromosomes and suppress spindle formation in other parts of the ooplasm. This spatially restricted spindle assembly requires the local activation of key proteins important for bipolar spindle assembly near the chromosomes. This spatial regulation is not only limited to microtubule nucleation factors but also to other regulators of microtubule dynamics and organisation, because non-spindle microtubules are present all over the oocyte yet fail to organise into a spindle. Chromosomes serve as spatial cues, which can be sensed by key regulators of spindle assembly. These proteins can then be locally activated to execute their functions, together with many other ubiquitous proteins whose activity is not spatially regulated.

Institute of Cell Biology, School of Biological Sciences, University of Edinburgh, Edinburgh EH9 3BF, UK.

*Author for correspondence (h.ohkura@ed.ac.uk)

 H.O., 0000-0003-4059-431X

The Ran–Importin pathway is the most-studied system for spatial regulation in oocytes. Ran is a small GTPase with two forms – active RanGTP and inactive RanGDP. RanGDP is converted into RanGTP by chromatin-bound RCC1, a guanine nucleotide exchange factor (GEF) (Renault et al., 2001). Cytoplasmic Ran GTPase-activating protein (RanGAP) promotes GTP hydrolysis into GDP (Seewald et al., 2002). Owing to chromosomal localisation of RCC1, the RanGTP concentration is higher around the chromosomes, generating a RanGTP gradient (Kalab et al., 2002). Proteins important for spindle assembly are inhibited by Importin binding, and RanGTP releases them from Importin for local activation near the chromosomes (Gruss et al., 2001; Nachury et al., 2001). These proteins are collectively called 'spindle assembly factors'. So far, over 20 spindle assembly factors have been identified (Cavazza and Vernos, 2015).

Among the spindle assembly factors, TPX2 is one of the first and most-studied factors. TPX2 plays multifaceted roles in spindle assembly in meiosis. After RanGTP releases TPX2 from importin inhibition, TPX2 interacts with Aurora A to activate it and with Kinesin-5 to modulate its distribution at the spindle. TPX2 also forms condensates on microtubule lattices and recruits other microtubule nucleators, Augmin and γ-TuRC, for branching microtubule nucleation (Alfaro-Aco et al., 2017, 2020; King and Petry, 2020; Petry et al., 2013). TPX2 plays crucial roles in spindle microtubule assembly, pole focusing and size control in mouse oocytes and *Xenopus* egg extracts (Brunet et al., 2008; Helmke and Heald, 2014).

In *Xenopus* egg extracts, a dominant negative Ran prevents spindle microtubule assembly, whereas hyperactive RanQ69L can induce spindle-like structures in the absence of chromosomes (Carazo-Salas et al., 1999; Kalab et al., 1999; Ohba et al., 1999; Wilde and Zheng, 1999). However, in mouse oocytes, abolishing the RanGTP gradient either by expressing dominant negative or hyperactive Ran leads to defects in the meiosis I spindle without disrupting the spatially restricted spindle assembly (Dumont et al., 2007). Similarly, in *Drosophila* oocytes, these dominant mutants affect spindle organisation, but a spindle is still formed only around the chromosomes (Cesario and McKim, 2011). In human oocytes, expressing dominant negative Ran severely delays an impaired spindle assembly around the chromosomes (Holubcová et al., 2015). Therefore, these observations suggest a varied importance of the Ran pathway in different species, but also the presence of alternative mechanisms for the spatial regulation of proteins important for bipolar spindle assembly in oocytes.

The chromosomal passenger complex (CPC) containing Aurora B kinase has been proposed as an alternative signal for spatial regulation in oocytes. The CPC is essential for spindle microtubule assembly in *Xenopus* egg extract and in *Drosophila* oocytes (Colombié et al., 2008; Radford et al., 2012; Sampath et al., 2004). The CPC is activated on the chromosomes independently of Ran (Kelly et al., 2007), and directly phosphorylates and inactivates

microtubule-destabilising proteins, such as MCAK (also known as KIF2C) (Andrews et al., 2004; Lan et al., 2004; Ohi et al., 2004) and Op18 (also known as Stathmin) (Gadea and Ruderman, 2006). In addition, the CPC co-operates with a 14-3-3 family phospho-docking protein to locally activate spindle-associated proteins around the chromosomes in *Drosophila* oocytes. 14-3-3 binds to the Kinesin-14 protein Ncd to prevent it from binding to non-spindle microtubules away from the chromosomes. Chromosome-bound CPC phosphorylates an additional site on Ncd to release it from inhibition by 14-3-3 (Beaven et al., 2017). Many more spindle-associated proteins have been found to be regulated by 14-3-3, including the CPC and the centralspindlin complex (Repton et al., 2022).

RCC1 and the CPC are two chromosomal cues so far identified for the spatial regulation of spindle-associated proteins in oocytes. However, it is unknown whether they can account for all of the spatial regulation in oocytes, as no systematic studies have been done to identify spatially regulated spindle proteins, chromosomal signals or regulatory mechanisms. In this report, we developed a novel method to assess whether the binding of a protein to microtubules is spatially regulated by inducing ectopic microtubules in *Drosophila* oocytes. Our findings on the *Drosophila* TPX2 homologue Mei-38 suggest that a protein phosphatase-based mechanism spatially regulates this spindle-associated protein.

## RESULTS

### Induction of ectopic microtubules reveals spatial regulation of Mei-38 binding to microtubules in *Drosophila* oocytes

To identify the spindle-associated proteins that are locally activated near the chromosomes and bind to microtubules, we induced ectopic microtubule clusters by treating *Drosophila melanogaster* oocytes with the microtubule-stabilising drug taxol (paclitaxel) (Fig. 1A). After taxol treatment, we immunostained wild-type mature oocytes using antibodies against various spindle-associated proteins or mature oocytes expressing GFP-tagged spindle-associated proteins using an anti-GFP antibody, together with an anti-α-tubulin antibody and DAPI.

Without taxol treatment, mature *Drosophila* oocytes naturally arrest in metaphase I with a bipolar spindle. After a brief taxol treatment, the bipolar spindle associated with the chromosomes was still recognisable, although the morphology was often distorted to various degrees. In addition, many ectopic microtubule clusters were formed all over the oocytes (Fig. 1A). The intensity and morphology varied, but the maximum signal intensity of some clusters was comparable to that of the spindle microtubules. For the localisation of spindle-associated proteins, we classified the outcomes into two types depending on whether they are spatially regulated (Fig. 1A): (1) if a spindle-associated protein indiscriminately binds to microtubules, it would equally localise to the entire spindle and ectopic microtubule clusters; (2) if a protein has a higher affinity to microtubules near the chromosomes than further away, a higher concentration of the protein would be observed on the spindle microtubules near chromosomes than ectopic microtubules with a similar microtubule density.

Among 12 spindle proteins that we tested, three showed equal localisation to all microtubules (Fig. 1A; Fig. S1), indicating that they are unregulated (type 1). They included TACC, the Kinesin-13 Klp10A and the Kinesin-5 Klp61F. For example, TACC showed uniform distribution on the spindle microtubules and ectopic microtubules, reflecting the intensity and pattern of tubulin signals (Fig. 1B). To quantify the signal intensity, a line was drawn along the long axis of the spindle and another line was drawn over a microtubule cluster with roughly the same tubulin intensity as the

spindle microtubules (Fig. 1C). The intensity of the signal was measured along the lines and normalised using the maximum intensity of the spindle microtubules in each oocyte. The mean signal intensity of TACC roughly followed the tubulin mean intensity along the spindle axis and also along the line drawn across ectopic microtubule clusters (Fig. 1C). These proteins are likely to bind to microtubules indiscriminately under this condition. In other words, they are not regulated spatially in terms of microtubule binding.

In contrast, the other nine proteins showed stronger signals on the spindle microtubules, especially near the chromosomes, than ectopic microtubules indicating that they are spatially regulated (type 2). They include Mars (the ortholog of HURP), the CPC subunits, the Kinesin-14 Ncd, Mink (the ortholog of NuSAP), Cyclin B and Mei-38 (the ortholog of TPX-2) (Fig. 1A, Figs S2, S3, Fig. 2B). For example, the signal of the GFP-tagged Mars was much stronger on spindle microtubules than on ectopic microtubule clusters, even where the tubulin intensity was comparable (Fig. 1D). Furthermore, the GFP–Mars signal was stronger near the chromosomes, and became weaker away from the chromosomes even within the same spindle (Fig. 1D). Quantification showed that the GFP–Mars signal intensity on the spindle microtubules decayed much more sharply further away from the chromosomes than the tubulin intensity. Moreover, the GFP–Mars signal on ectopic microtubules was comparable to the background without a distinct peak (Fig. 1E). Therefore, Mars binding to microtubules is strongest near the chromosomes, becomes progressively weaker further away along the spindle and is virtually absent on ectopic microtubules, indicating that Mars is spatially regulated in terms of microtubule binding.

### Truncation analysis narrows the region sufficient for spatial regulation

We focused our study on Mei-38, the TPX2 homologue, among the spatially regulated proteins we identified, as it apparently lacks sequences potentially regulated by previously known mechanisms, such as importin-binding sites and 14-3-3-binding sites. Mammalian TPX2 is regulated by Ran-importin through nuclear localisation signals (Gruss et al., 2001) but Mei-38 lacks nuclear localisation signals, as well as the Aurora A and Kinesin-5 interaction domains found in mammalian TPX2 (Goshima, 2011; Fig. 2A).

In taxol-treated oocytes, Mei-38 localised to the spindle microtubules much more strongly than to ectopic microtubule clusters, even where the tubulin intensity was comparable (Fig. 2B). Furthermore, within the same spindle, the Mei-38 signal was stronger near the chromosomes and became gradually weaker towards the poles (Fig. 2B). Quantification showed that the Mei-38 signal intensity on the spindle microtubules decayed more sharply further away from the chromosomes than the tubulin intensity. Moreover, the Mei-38 signal on ectopic microtubules was close to the background with a very small peak (Fig. 2B).

To define domains responsible for spindle binding and its spatial regulation, we carried out a taxol assay in oocytes expressing various GFP-tagged truncated Mei-38 proteins (Fig. 2A,B; Fig. S5). Goshima (2011) has identified three regions conserved among the TPX2 family across species, including in humans. Mei-38(1–278), Mei-38(95–323) and Mei-38(95–278), lacking the conserved region 1, 3 or both, still maintained spatial regulation identical to the full-length protein, although the signal intensity was lower than that for the full-length Mei-38 (Fig. 2A,C; Fig. S5). This demonstrates that these regions are not essential for spindle localisation or spatial regulation.

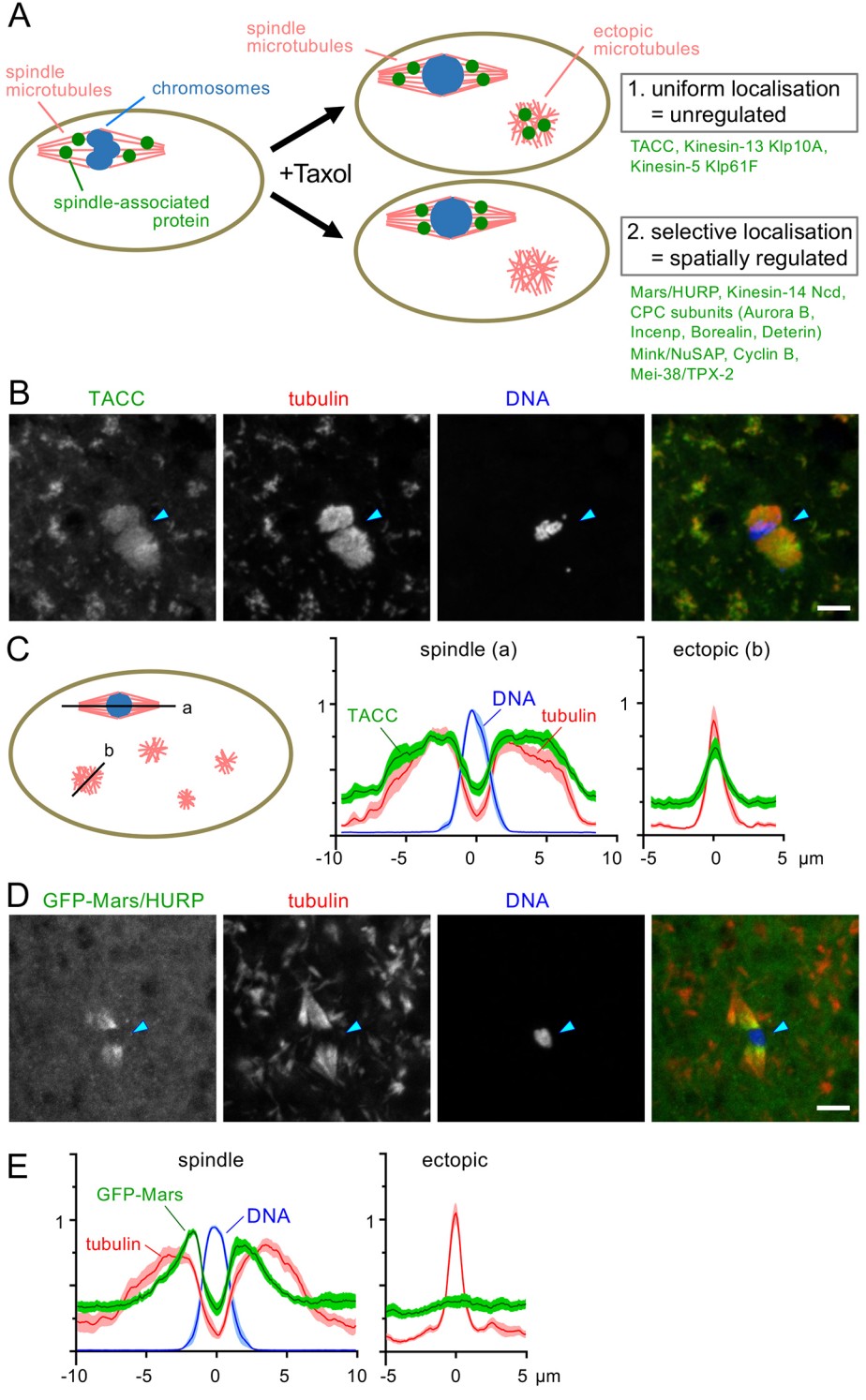

**Fig. 1. Induction of ectopic microtubules identified spindle-associated proteins that are spatially regulated in *Drosophila* oocytes.** (A) Induction of ectopic microtubules with taxol reveals two types of spindle-associated proteins. Proteins that localise uniformly to spindle and ectopic microtubules are not spatially regulated in terms of microtubule binding. In contrast, proteins that selectively localise to spindle microtubules are spatially regulated. (B) TACC localisation in taxol-treated oocytes. Wild-type mature oocytes, which are naturally arrested in metaphase I, were incubated with taxol and immunostained using anti-TACC and anti-$\alpha$-tubulin antibodies. TACC uniformly localised to both spindle and ectopic microtubules. Arrowheads indicate chromosomes. Scale bar: 5 µm. (C) Quantification of TACC, $\alpha$-tubulin and DNA signal intensities. The first line (a) was drawn along the long axis of the spindle microtubules. The second line (b) was drawn on an ectopic microtubule cluster with a similar maximum intensity to the spindle microtubules. The centre of the chromosomes or the ectopic cluster is defined as 0. Signal intensities were measured along the lines and normalised to the maximum intensity value of the spindle line in each oocyte. The graphs show means±s.e.m. (*n*=11). (D) GFP–Mars localisation in taxol-treated oocytes. Mature oocytes expressing GFP–Mars were incubated with taxol and immunostained using anti-GFP and anti-$\alpha$-tubulin antibodies. GFP–Mars localised specifically to the spindle microtubules, not to ectopic microtubules. Within the spindle microtubules, the signal is stronger near the chromosomes. Arrowheads indicate chromosomes. Scale bar: 5 µm. (E) Quantification of GFP-Mars, $\alpha$-tubulin and DNA signal intensities following the same method as C (*n*=11).

Mei-38(143–278) and Mei-38(176–278) localised uniformly to spindle microtubules but much more weakly to ectopic microtubules (Fig. 2A–C; Fig. S5), suggesting that the area between the conserved regions 1 and 2 contributes to spatial regulation to some degree. Mei-38(1–221) lacking the conserved regions 2 and 3 failed to localise to any microtubules (Fig. 2A,C; Fig. S5), suggesting the conserved region 2 and the C-terminal flanking area are important for microtubule association. We have identified Mei-38(176–278) containing the conserved region 2 as

the shortest fragment sufficient to localise to spindle microtubules. This fragment is still under spatial regulation, as it showed much stronger localisation to the spindle microtubules than to ectopic ones (Fig. 2B). A smaller fragment, Mei-38(176–242), failed to localise to any microtubules. A western blot confirmed that this fragment was expressed at a similar level to the full-length or other truncated proteins (Fig. S6). Mei-38(195–323) failed to localise to any microtubules (Fig. S6), probably because of a very low protein level seen in a western blot (Fig. S6).

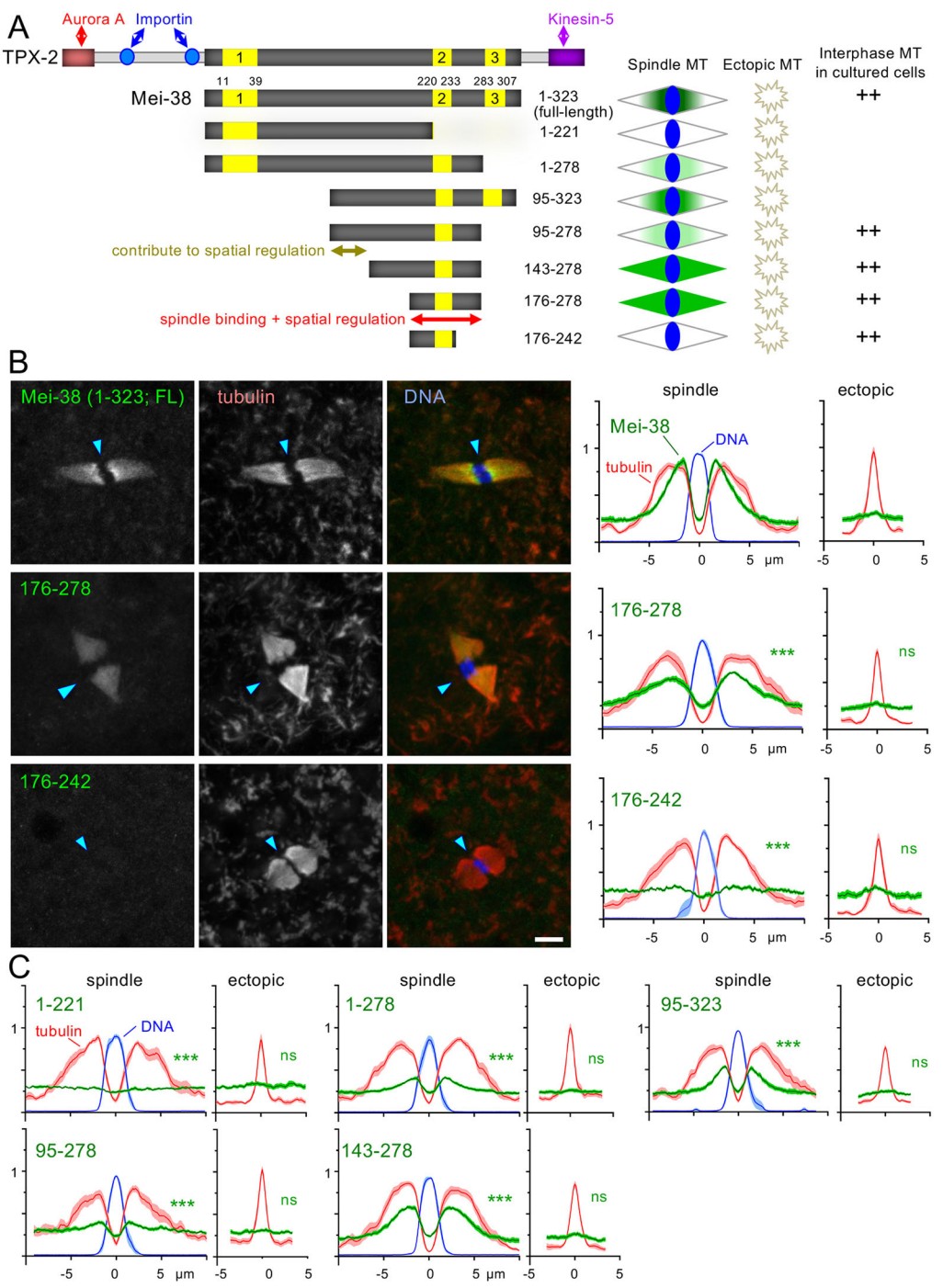

**Fig. 2. Truncations of Mei-38 defined a spindle-binding domain that is regulated spatially.** (A) A summary diagram of Mei-38 truncations and their localisation in taxol-treated oocytes and interphase S2 cells. Numbers indicate amino acid residues. The region including the conserved domain 2 is important for microtubule binding and spatial regulation. (B) Localisation of GFP-tagged Mei-38 truncations in taxol-treated oocytes. The arrowheads indicate the position of the chromosomes. Quantifications were carried out as in Fig. 1C. For comparison, the profile of the Mei-38 mean signal intensity of each truncation is presented after normalisation to adjust the mean background signal intensity (at 0 in the spindle line) to be identical to the control (the full-length Mei-38). Scale bar: 5 µm. ***$P<0.001$; ns, not significant ($P>0.05$) [two-tailed unpaired $t$-tests when the signal intensity of a GFP–Mei-38 variant is compared to that of GFP–Mei-38(1–323; FL)]. $n$=14, 14, 11 for Mei-38 fragments 1–323 (FL), 176–278 and 176–242, respectively. $P$=4.5×10$^{-8}$, 7.6×10$^{-12}$ (spindle) and $P$=0.83, 0.33 (ectopic) for 176–278 and 176–242, respectively. (C) Quantification of the remaining truncations in B. Images are shown in Fig. S5. $n$=12, 14, 13, 13, 12; $P$=5.6×10$^{-13}$, 4.4×10$^{-10}$, 2.4×10$^{-11}$, 3.3×10$^{-10}$, 4.5×10$^{-8}$ (spindle) and $P$=0.16, 0.38, 0.11, 0.50, 0.33 (ectopic) for Mei-38 fragments 1–221, 1–278, 95–323, 95–278 and 143–278, respectively. The spreads of the individual data points used for statistical analysis are shown in Fig. S10.

In summary, we narrowed down the region sufficient for the spindle localisation and spatial regulation in oocytes, although another region also contributes to spatial regulation.

### Identification of a microtubule-binding domain containing a highly conserved sequence

To assess the microtubule-binding activity separately from oocyte-specific regulation, we took advantage of a *Drosophila* embryonic S2 cultured cell line. A previous report (Goshima, 2011) has shown that GFP–Mei-38 expressed in S2 cells is associated with filamentous microtubule networks in interphase, and assessed the microtubule binding activity of some truncated proteins. We found that fragments that bind to spindle microtubules in oocytes, including Mei-38(176–278), were able to associate with interphase microtubules in S2 cells (Fig. 3A). Unexpectedly, Mei-38(176–242), which failed to localise to any microtubules in oocytes, was able to associate with interphase microtubules in S2 cells (Fig. 3A). Therefore, this deleted sequence 243–278 is dispensable for microtubule binding *in vitro* and in S2 cells but is required for spindle microtubule localisation in oocytes.

Mei-38(176–242) contains region 2, which is also highly conserved in vertebrates (Goshima, 2011). To test whether Mei-38(176–242) has intrinsic microtubule binding activity *in vitro*, we produced the fragment in bacteria as a fusion with MBP (Fig. S7). Pure porcine tubulin dimers were incubated with bacterially produced Mei-38(176–242) and were polymerised into

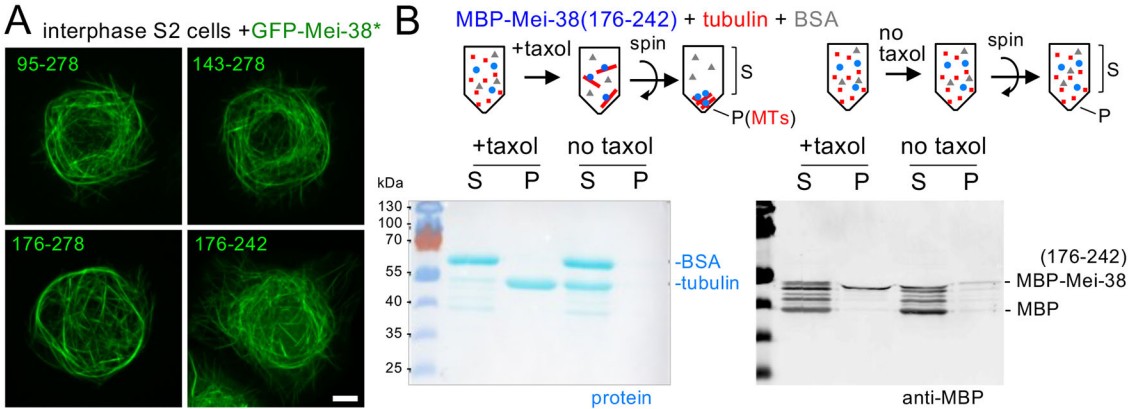

**Fig. 3. Mei-38(176–242) binds to microtubules in interphase S2 cells and *in vitro*.** (A) Interphase S2 cells expressing GFP-tagged truncated Mei-38 is known to associate with interphase microtubules. Mei-38(176–242), which does not localise to any microtubules in oocytes binds to microtubules in interphase S2 cells. The experiment was done once, but more than 10 transfected cells each showed similar filamentous localisation. Scale bar: 5 μm. (B) Microtubule-binding assay of Mei-38(176–242). Purified MBP-tagged Mei-38(176–242) expressed in bacteria was mixed with tubulin and BSA. Taxol and GTP were added to one sample (+taxol), but not to the other sample (no taxol). Microtubules (MTs) were polymerised in the sample with taxol, both samples were centrifuged. The supernatants (S) and pellets (P) were analysed by protein staining and western blotting using an MBP antibody. Twice the amount of the pellets was loaded compared to the supernatants in each lane. The experiment was done once.

microtubules by taxol addition. After microtubules were spun down, the microtubule fraction (pellet) and non-binding fraction (supernatant) were analysed by western blotting using an anti-MBP antibody (Fig. 3B). MBP–Mei-38(176–242) was found in the microtubule fraction, whereas degradation products including the one with the size corresponding to MBP were missing (Fig. 3B). In agreement with S2 cells, this showed that the recombinant Mei-38(176–242) has an intrinsic microtubule-binding activity *in vitro*. In contrast, it fails to localise to the spindle or ectopic microtubules in oocytes.

### S225 is responsible for spatial regulation of the Mei-38(176–278) binding to microtubules

We then decided to focus on Mei-38(176–278), the smallest fragment that localises to the spindle in oocytes (Fig. 4A), because it is still spatially regulated. This fragment shows significantly stronger localisation to the spindle microtubules compared to ectopic microtubules in taxol-treated oocytes (Fig. 4C,D). It contains three sequences of interest: (1) a central highly conserved region containing the potential phosphorylation site S225 (Fig. 4A,B), (2) the N-terminal flanking region 198–203, which matches the L/F/xxI/VxE docking motif for PP2A-B56 (which has two isoforms in *Drosophila* encoded by *wdb* and *wrd*) (Hertz et al., 2016) (Fig. 4A), and (3) the C-terminal flanking sequence 243–278, which is dispensable for microtubule-binding *in vitro* or in S2 cells but is required for spindle localisation in oocytes (Fig. 4A).

We tested the role of S225 in binding of Mei-38(176–278) to spindle and ectopic microtubules in taxol-treated oocytes by mutating it to alanine (S225A) (Fig. 4B,C). Strikingly, this mutated fragment, Mei-38(176–278,S225A), localised to all microtubules equally including ectopic microtubules, completely abolishing the spatial regulation of microtubule binding (Fig. 4C,D). This result demonstrated that S225 is solely responsible for spatial regulation of this fragment.

### PP2A-B56 docking enhances Mei-38 localisation to spindle microtubules

Next, we looked into the role of a flanking PP2A-B56-docking motif located near S225. This sequence is conserved among *Drosophila* species. To test its role, we mutated three consensus residues of this motif to alanine residues (B56-3A) in Mei-38(176–278) (Fig. 4A,C). This mutation greatly reduced the localisation of Mei-38(176–278) to the spindle microtubules (Fig. 5A). This suggests that a failure to dephosphorylate this fragment by PP2A-B56 suppresses the localisation to spindle microtubules. To test the involvement of S225 in this process, we further introduced a non-phosphorylatable mutation, S225A, to generate Mei-38(176–278,B56-3A,S225A). Spindle microtubule localisation was restored by S225A in taxol-treated oocytes (Fig. 5A), suggesting that dephosphorylation of S225 by PP2A-B56 is important for spindle microtubule localisation.

We then looked into the role of the C-terminal flanking sequence 243–278. Mei-38(176–242), which lacks this sequence 243–287 has intrinsic microtubule-binding activity but fails to localise to the spindle in oocytes (Fig. 5B). Addition of the sequence 243–278 enabled spindle localisation in oocytes. We hypothesised that the sequence 243–278 promotes dephosphorylation of Mei-38 on the spindle, and without this sequence, Mei-38(176–242) remains fully phosphorylated in oocytes even near the chromosomes and spindle, preventing its binding to both spindle and ectopic microtubules. To test this hypothesis, we mutated S225 of this fragment to alanine. As predicted, this mutant fragment, Mei-38(176–242,S225A), localised to the spindle microtubules in taxol-treated oocytes, although weakly (Fig. 5B). Therefore, this fragment, Mei-38(176–242), failed to localise to the spindle microtubules in taxol-treated oocytes, partially because the sequence 243–278 is important for dephosphorylation at S225, which normally takes place in the spindle area.

The S225A mutation abolished spatial regulation of Mei-38(176–278), allowing it to localise equally to both spindle and ectopic microtubules. In contrast, introduction of S225A to either Mei-38(176–278, B56-3A) or Mei-38(176–242) enhanced the localisation to the spindle microtubules, but the localisation to ectopic microtubules was weaker than the spindle (Fig. 5A,B). This indicates that, although S225 is solely responsible for spatial regulation of Mei-38(176–278), there is a cryptic mechanism that spatially regulates Mei-38 in the absence of the PP2A-B56-docking site or the C-terminal region 243–278.

Finally, we tested the importance of S225 and the PP2A-B56-docking motif for the localisation of the full-length Mei-38. We

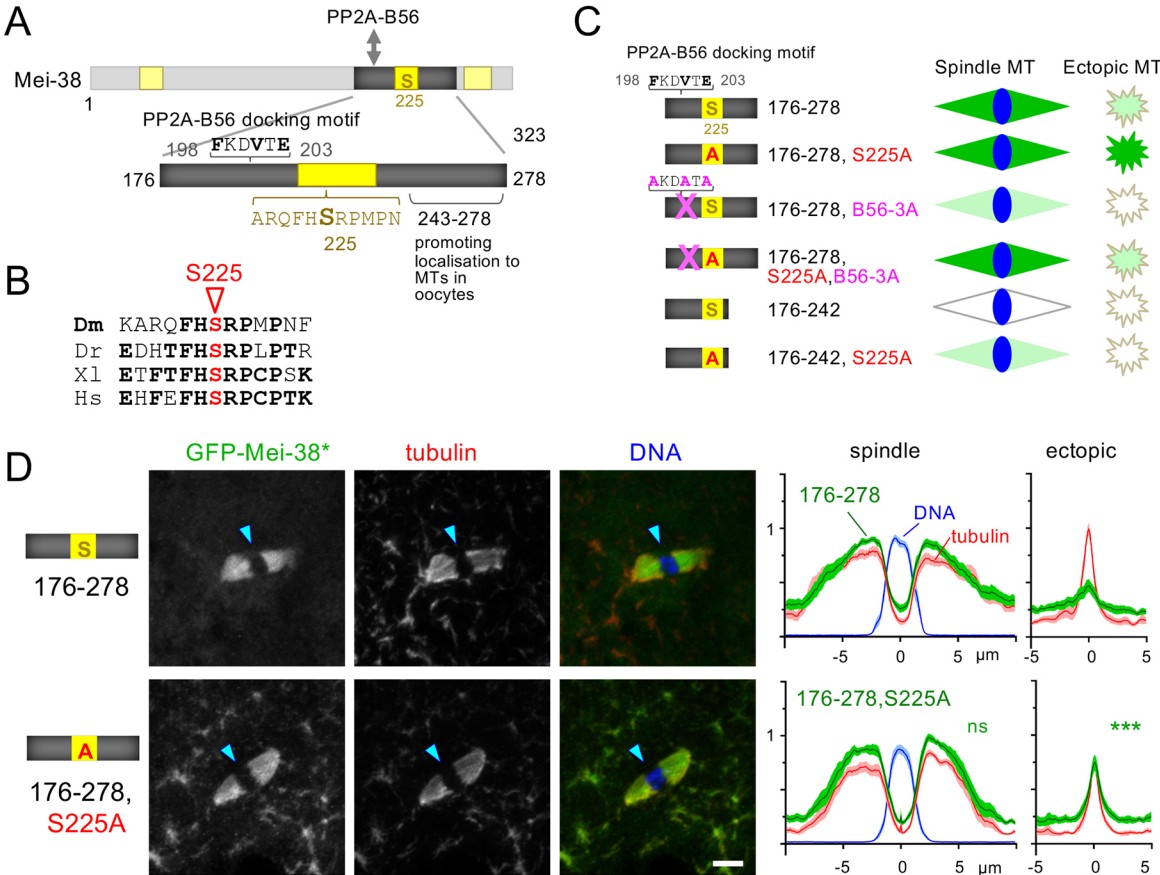

**Fig. 4. The non-phosphorylatable mutation S225A abolished spatial regulation of Mei-38(176–278).** (A) The minimal fragment Mei-38(176–278) under spatial regulation contains three regions of interest – a PP2A-B56-docking motif, the conserved region 2 with a potential phosphorylation site (S225) and a region (243–278) that is dispensable for microtubule binding but important for spindle association in oocytes. (B) The conserved region 2 containing S225 is conserved also in vertebrates. Dm, *Drosophila melanogaster*; Dr, *Danio rerio*; Xl, *Xenopus laevis*; Hs, *Homo sapiens*. (C) A summary diagram of various mutants and truncations within the minimum fragment under the control of spatial regulation (176–278) with the localisation in taxol-treated oocytes shown in D and Fig. 5. B56-3A, three key residues of the PP2A-B56-docking motif were mutated to alanine; S225A, a non-phosphorylatable mutation of S225 within the conserved region 2; 176–242, a region that can bind to microtubules in S2 cells and *in vitro*, but cannot localise to spindle microtubules in oocytes. (D) GFP–Mei-38(176–278) and GFP–Mei-38(176–278,S225A) localisation in taxol-treated oocytes. A non-phosphorylatable mutation at S225 (S225A) allowed a GFP–Mei-38(176–278) fragment to localise to all microtubules equally, abolishing the spatial regulation. The arrowheads indicate the position of the chromosomes. Scale bar: 5 µm. The signal distribution was quantified as Fig. 2B. ***$P<0.001$; ns, not significant ($P>0.05$) (two-tailed unpaired *t*-tests when the signal intensity of a GFP–Mei-38 variant is compared to that of 176–278). $n=10$, 12 for fragments 176–278 and 176–278,S225A, respectively. $P=0.69$ (spindle) and $P=4.0\times10^{-6}$ (ectopic). The spreads of the individual data points used for statistical analysis are shown in Fig. S10. For comparison, the profile of the Mei-38 mean signal intensity of each truncation is presented after normalisation to adjust the mean background signal intensity (at 0 in the spindle line) to be identical to that for Mei-38(176–278). Note that a microscope different from the one for Figs 1 and 2 was used to acquire the data in this figure and the following figures, and gave a lower background.

first expressed the GFP-tagged full-length Mei-38 with non-phosphorylatable or phospho-mimetic mutations at S225, Mei-38(S225A) and Mei-38(S225D), in oocytes (Fig. 6A). The distribution pattern of the non-phosphorylatable mutant, Mei-38(S225A), was similar to Mei-38 without the mutation in taxol-treated oocytes, although the signal appeared to be weaker (Fig. 6A,B). By contrast, the phospho-mimetic mutant Mei-38(S225D) failed to localise to either spindle or ectopic microtubules (Fig. 6A,B). This suggests that single phosphorylation of S225 is sufficient to prevent the full-length Mei-38 from binding to microtubules, but the full-length Mei-38 also has another mechanism that suppresses microtubule binding away from the chromosomes, which is consistent with our truncation analysis. Next, to test whether PP2A-B56 docking is important for the localisation of the full-length Mei-38, we expressed the GFP-tagged full-length Mei-38 with the mutated PP2A-B56-docking site, Mei-38(B56-3A), in oocytes. The distribution of the Mei-38(B56-3A)

localisation in taxol-treated oocytes was similar to Mei-38 without the mutation, but the signal was generally weaker (Fig. 6A,C). This docking site might be less important in the context of the full-length Mei-38.

To confirm that PP2A-B56 regulates Mei-38 in oocytes, we knocked down each of two PP2A-B56 isoforms (*wdb*, *wrd*) in oocytes expressing GFP–Mei-38 by co-expressing shRNA against them (Fig. S8). Single knockdown of *wrd* did not significantly change the Mei-38 localisation to the spindle, whereas single knockdown of *wdb* increased it. To minimise effects of functional redundancy, double knockdown of *wdb* and *wrd* was carried out, which showed a significant decrease in the GFP–Mei-38 localisation to the spindle. Although this decreased localisation in the double knockdown is consistent with the effect of the mutation in the B56-docking site, it should be noted that these changes were relatively small, the RNAi efficiencies are unknown and PP2A-B56 has many other substrates.

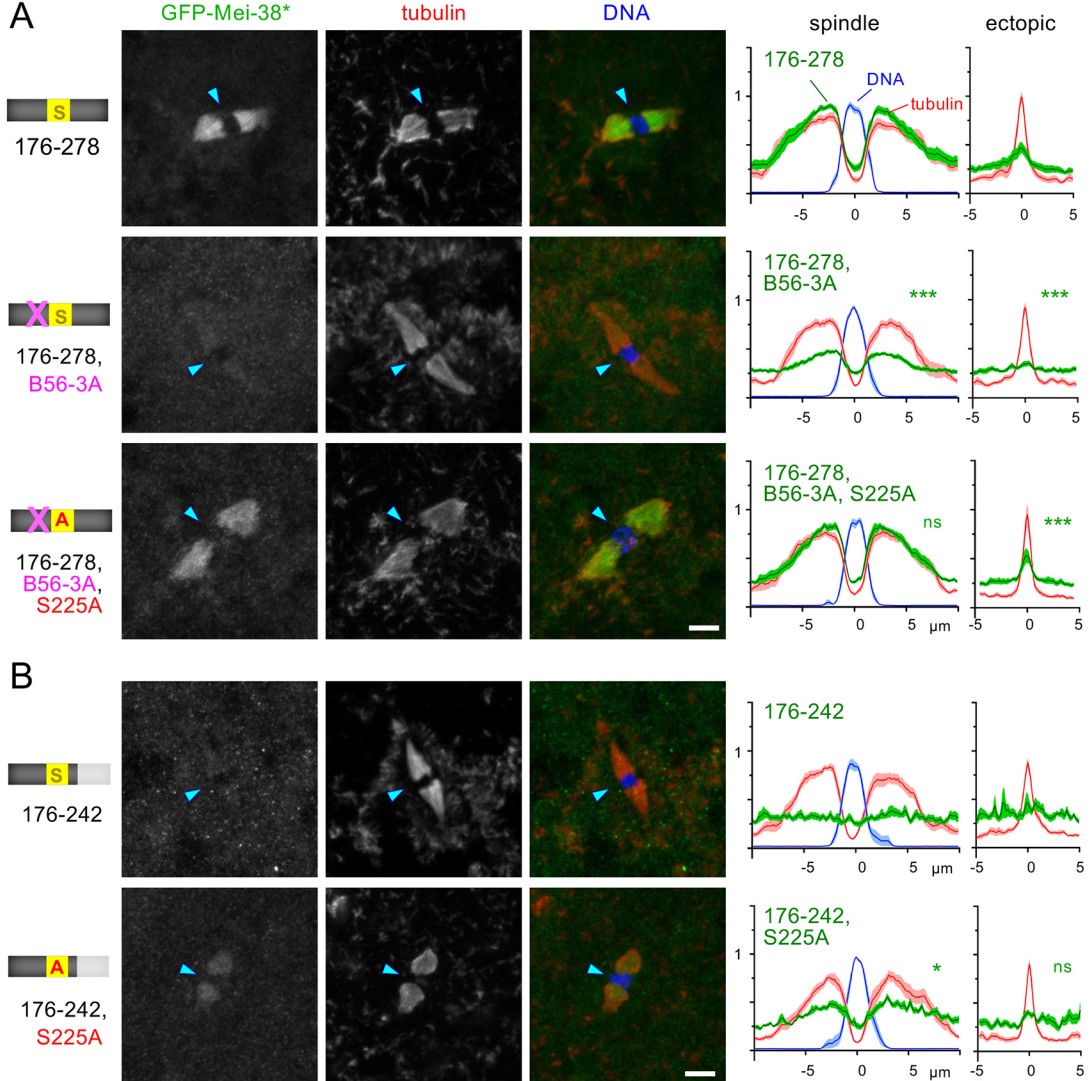

**Fig. 5. Dephosphorylation of S225 by PP2A-B56 might promote spindle microtubule binding of Mei-38.** (A) Localisation of GFP–Mei-38(176-278) carrying mutations in the PP2A-B56-docking motif (B56-3A) and/or a non-phosphorylatable mutation at S225 (S225A) in taxol-treated oocytes. The arrowheads indicate the position of the chromosomes. Scale bar: 5 µm. The signal distribution was quantified as Fig. 2B. The panel for GFP–Mei-38(176-278) is identical to Fig. 4D, and is shown for comparison. ***$P$<0.001; ns, not significant ($P$>0.05) (two-tailed unpaired $t$-tests when the signal intensity of a GFP–Mei-38 variant is compared to that of fragment 176–278). $n$=10, 13, 13 for fragments 176–278, 176–278,B56-3A and 176–278,B56-3A,S225A, respectively. $P$=8.7×10$^{-12}$, 0.088 (spindle) and $P$=9.77×10$^{-4}$, 8.7×10$^{-4}$ (ectopic) for 176–278,B56-3A and 176–278,B56-3A,S225A, respectively. (B) Localisation of GFP–Mei-38(176–242) with or without a non-phosphorylatable mutation at S225 (S225A) in taxol-treated oocytes. The arrowheads indicate the position of the chromosomes. Scale bar: 5 µm. The signal distribution was quantified as Fig. 2B. *$P$<0.05; ns, not significant ($P$>0.05) [two-tailed unpaired $t$-tests when the signal intensity of GFP–Mei-38(176–242,S225A) is compared to that of fragment 176–242]. $n$=12, 12 for fragments 176–242 and 176–242,S225A, respectively]. $P$=0.030 (spindle) and 0.57 (ectopic). The spreads of the individual data points used for statistical analysis are shown in Fig. S10.

## DISCUSSION

The bipolar spindle forms only around the meiotic chromosomes in oocytes without centrosomes in the exceptionally large cytoplasm. This requires key proteins to be locally activated in oocytes. In this study, we have developed a novel method to assess spatial regulation of specific proteins in oocytes by inducing ectopic microtubules using taxol. We have identified nine spindle-associated proteins that are spatially regulated in *Drosophila* oocytes in terms of microtubule binding. Our analysis of the *Drosophila* TPX2 homologue Mei-38 suggests a phosphatase-driven mechanism, in which inhibitory phosphorylation of Mei-38 is removed by PP2A-B56, concentrated at kinetochores, enables it to bind to spindle microtubules (Fig. 6D).

In oocytes, spatially restricted activation of key proteins around the chromosomes is important for the following reasons. First, oocytes lack centrosomes, the main microtubule-organising centres in mitotic cells, and need to restrict a high level of microtubule nucleation and/or stabilisation to the proximity of the chromosomes. Second, non-spindle microtubules co-exist with spindle microtubules in oocytes but should not organise into bipolar spindles. Key microtubule-organising proteins, such as crosslinkers and motors, must be active only near the chromosomes. Third, the density of non-spindle microtubules in oocytes is low, but the volume of the oocyte is far larger than that of the spindle (>1000 times in *Drosophila* oocytes). A large number of non-spindle microtubules might outcompete spindle microtubules for binding of

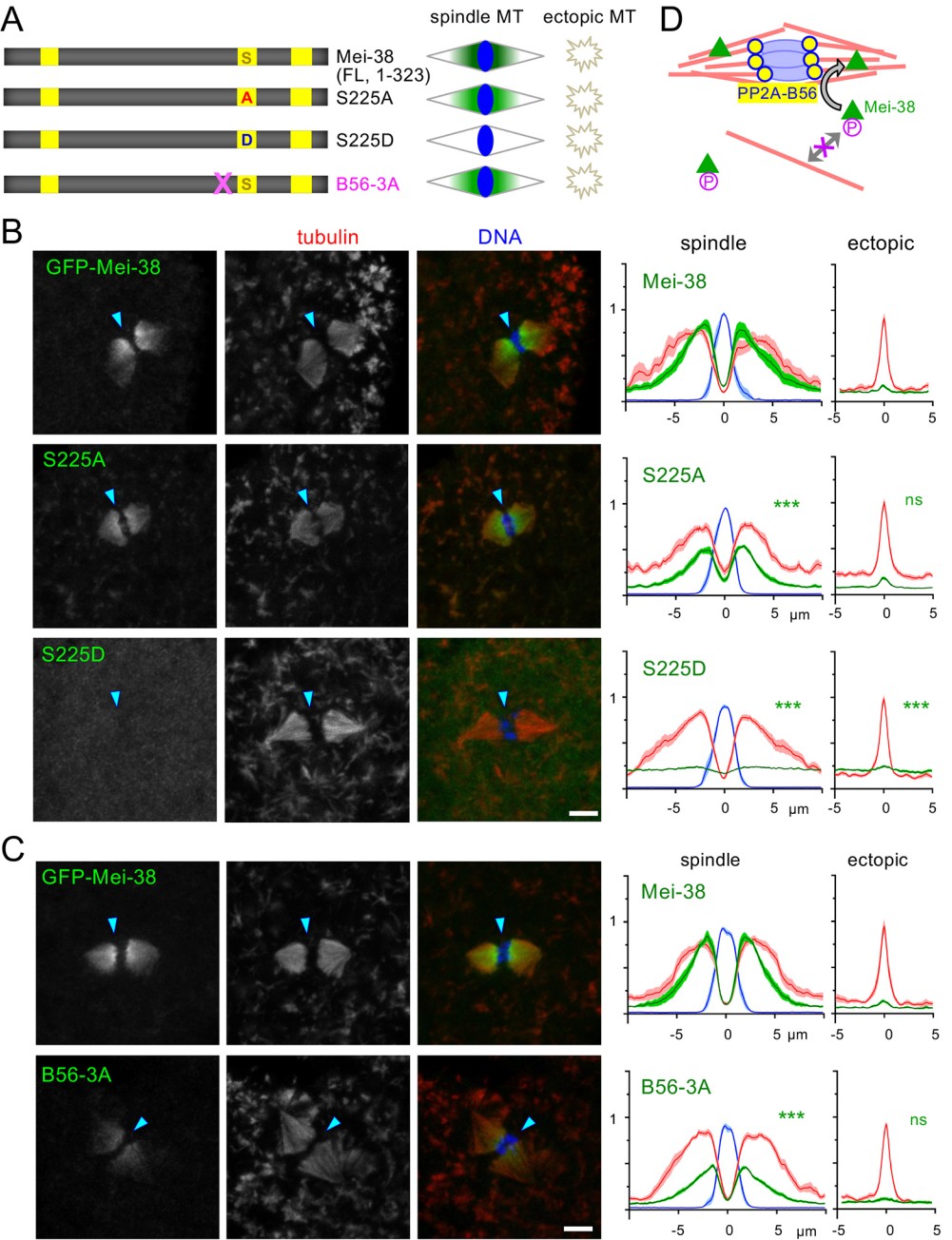

**Fig. 6. A phospho-mimetic mutation at S225 prevents Mei-38 from binding to microtubules in taxol-treated oocytes.** (A) A summary diagram of the GFP-tagged full-length Mei-38 with mutations in S225 and a PP2A-B56-docking motif, together with their localisation in taxol-treated oocytes. GFP–Mei-38(S225A) and GFP–Mei-38(S225D) carry non-phosphorylatable and phospho-mimetic residues (S225A and S225D), respectively. GFP–Mei-38(B56-3A) carries mutations at the PP2A-B56-docking motif. (B) Localisation of GFP–Mei-38, GFP–Mei-38(S225A) and GFP–Mei-38(S225D) in taxol-treated oocytes. The methods and quantification are the same as Fig. 2B. ***$P<0.001$; ns, not significant ($P>0.05$) (two-tailed unpaired $t$-tests when the signal intensity of a GFP–Mei-38 variant is compared to that of GFP–Mei-38). $n=9$, 14, 18 for Mei-38, S225A and S225D, respectively. $P=5.2\times10^{-6}$, $1.2\times10^{-15}$ (spindle) and $P=0.77$, $7.5\times10^{-4}$ (ectopic) for S225A and S225D, respectively. The spreads of the individual data points used for statistical analysis are shown in Fig. S10. These oocytes were processed and observed in parallel. The arrowheads indicate the position of the chromosomes. Scale bar: 5 µm. (C) Localisation of GFP–Mei-38 and GFP–Mei38(B56-3A) in taxol-treated oocytes. The methods and quantification are the same as Fig. 2B. ***$P<0.001$; ns, not significant ($P>0.05$) (two-tailed unpaired $t$-tests when the signal intensity of a GFP–Mei-38 variant is compared to that of GFP–Mei-38). $n=13$, 17 for Mei-38 and B56-3A, respectively. $P=5.9\times10^{-8}$ (spindle) and 0.088 (ectopic). The spreads of the individual data points used for statistical analysis are shown in Fig. S10. These oocytes were processed and observed in parallel. The arrowheads indicate the position of the chromosomes. Scale bar: 5 µm. (D) A proposed model in which the phosphatase PP2A-B56, concentrated at kinetochores, removes inhibitory phosphorylations from Mei-38, enabling it to bind to spindle microtubules.

key microtubule-associated proteins. To overcome this, oocytes need to either produce a far larger amount of the proteins than required or activate the microtubule-binding activity of these proteins only near the chromosomes.

Understanding the mechanisms governing spatial regulation of proteins important for spindle formation in oocytes still remains limited. For local activation of the spindle-associated proteins in oocytes, RCC1 (GEF) and the CPC (a kinase complex) are the two

chromosomal signals that have been identified so far. Further studies have identified the downstream pathways leading to local activation of key proteins, which involve Ran–Importin and 14-3-3, respectively (Beaven et al., 2017; Gruss et al., 2001; Nachury et al., 2001; Wiese et al., 2001). However, it is still unknown whether they account for all of the spatial regulation in oocytes, as no systematic studies have been carried out to identify chromosomal signals, spatially-regulated proteins or the underlying regulatory mechanisms.

Although proteins regulated by known mechanisms have been systematically identified (Clarke and Zhang, 2008; Kalab and Heald, 2008), these studies could not uncover an entirely new mechanism of spatial regulation. Instead, comprehensive understanding of spatially restricted spindle assembly in oocytes requires the systematic identification of spatially regulated proteins regardless of the mechanisms. In-depth studies of each spatially regulated protein could discover a new chromosomal signal and/or a signalling pathway, or add to a list of proteins controlled by a known mechanism. Studies of all spatially regulated proteins could lead to comprehensive understanding of chromosomal signals and the regulatory mechanisms.

Here, we developed a novel method to visualise a spatial distribution of microtubule binding affinity by inducing ectopic microtubule clusters in *Drosophila* oocytes. Using the method, we successfully identified nine spindle-associated proteins that are spatially regulated in terms of microtubule binding. A limitation of our method is that it only assesses general microtubule-binding activity. It does not assess interaction with dynamic microtubules or other activities, such as nucleation, cross-linking or motor activities. Furthermore, spatial regulation of some proteins might be indirect, as they could bind to other proteins that are spatially regulated or recognise spatially restricted signatures or structures. Nevertheless, our newly developed method is straightforward and rapid, and can be easily applied to other species to uncover conserved principles for spatial regulation in oocytes.

Among the spatially regulated proteins we identified, we focused our studies on the *Drosophila* TPX2 homologue Mei-38, as it appeared to lack sites potentially regulated by known mechanisms. Our truncation and mutational analysis has identified a microtubule-binding domain (176–278) in Mei-38 that contains a highly conserved sequence among the TPX2 family (Goshima, 2011). We demonstrated the microtubule-binding activity of this domain *in vitro*, in interphase cultured cells and in metaphase I oocytes. Previous studies have identified multiple microtubule-binding domains in vertebrate TPX2 (Brunet et al., 2004; Guo et al., 2023; Trieselmann et al., 2003; Zhang et al., 2017), including a recent comprehensive study, which tested all ten modules in TPX2 individually and in combinations both *in vitro* and in cells (Liang et al., 2025). They detected microtubule-binding activity in six modules, but not in the module R6, which possibly corresponds to the microtubule-binding domain we identified in Mei-38. However, it is uncertain which module in TPX2 corresponds to this domain of Mei-38, as the overall organisation and primary sequence differ significantly between the two. All modules in TPX2 might have a potential microtubule-binding activity with considerable redundancy, but the actual activity of each module could have changed during evolution.

Our analysis of Mei-38 revealed a mechanism for its spatial regulation possibly driven by dephosphorylation. The spindle-binding domain, Mei-38(176–278), is under spatial regulation in oocytes, showing much stronger localisation to the spindle microtubules than to ectopic microtubules. A phospho-mimetic mutation at S225 prevented binding to any microtubules, whereas a non-phosphorylatable

mutation abolished the spatial regulation, resulting in indiscriminate binding to both spindle and ectopic microtubules. These results suggest that phosphorylation at S225 inhibits microtubule binding, and that this inhibitory phosphorylation is removed exclusively in the spindle area to allow binding to spindle microtubules.

Our results further suggest that the protein phosphatase PP2A-B56 promotes spindle microtubule binding of Mei-38(178–278) by targeting S225 through the adjacent PP2A-B56-docking motif. Additionally, our results also suggest the involvement of another flanking region in dephosphorylation at S225, and the presence of a cryptic mechanism which becomes evident only in the absence of the PP2A-B56-docking site or the flanking region. Furthermore, another region outside of 178–278 also contributes to spatial regulation, suggesting multiple layers of regulation on Mei-38.

As one of the two isoforms of PP2A-B56 is concentrated at the kinetochores in *Drosophila* oocytes (Jang et al., 2021), we propose that this phosphatase serves as a new chromosomal signal that locally activates proteins important for spindle formation (Fig. 6D). In oocytes, studies on PP2A-B56 have primarily focused on its regulation of kinetochore–microtubule attachment (Foley and Kapoor, 2013; Foley et al., 2011; Funabiki and Wynne, 2013; Suijkerbuijk et al., 2012). A protein phosphatase represents a new type of chromosomal signal, distinct from the previously known signals, a kinase (the CPC) and a GEF (RCC1). As inhibitory phosphorylation is a common way to block protein–protein interactions, including microtubule binding (Bramblett et al., 1993; Drewes et al., 1993; Hoshi et al., 1988; Nakajima et al., 2011; Shiina and Tsukita, 1999; Welburn et al., 2010), phosphatase-driven mechanisms might be widely used to locally activate more spindle-associated proteins near the chromosomes in oocytes. Furthermore, considering that PP2A-B56 is also concentrated at the kinetochores in mammalian oocytes (Yoshida et al., 2015), this phosphatase might act as a chromosomal signal in mammalian oocytes as well. It would be of great interest to test potential roles of PP2A-B56 or other phosphatases in mammalian oocytes.

## MATERIALS AND METHODS
### Molecular techniques
Standard molecular techniques were followed (Sambrook, 1989). Plasmids expressing Mei-38 or its variants were generated as below. A Gateway donor vector pENTR (Thermo Fisher Scientific) was linearised with NotI and AscI. Gene regions of interest flanked by a stop codon and overlapping regions to the ends of the linearised pENTR were amplified from a cDNA from the Nick Brown embryonic library (Brown and Kafatos, 1988) using PrimeSTAR DNA polymerase (Takara) or custom synthesised (IDT). This cDNA (pAG41) encodes a protein (Fig. S4) which is similar to Mei-38-polypeptide A but shorter by two residues. After purification, the linear vector and PCR products were ligated by Gibson assembly (HiFi; New England Biolabs). Sanger sequencing (Azenta) confirmed that no unwanted mutations were introduced during PCR. Gene regions of interest were then recombined into a Gateway destination vector using LR Clonase II enzyme (Thermo Fisher Scientific).

To introduce mutations of S225 (S225A and S225D) or the PP2A-B56-docking motif (B56-3A; from FKDVTE to AKDATA), upstream and downstream regions of a mutation were separately amplified using primers containing the mutation.

Two destination vectors were used for Gateway cloning. To express genes in flies under the UASp, φPGW was modified from the destination vector pPGW (https://flybase.org/reports/FBrf0179058) of the Murphy's Gateway collection by inserting the φC31 attB recombination sequence at the AatII site. To express genes under the Cu$^{2+}$-inducible metallothionein promoter in *Drosophila* S2 cells, the destination vector pMTGW (a gift from Gohta Goshima, Nagoya University, Japan) was used. To make transgenic flies, expression clones were microinjected into the fly line (BDSC9750;

PBac{y⁺-attP-3B}VK00033) using φC31 integrase-mediated transgenesis, which was carried out by BestGene Inc.

A double RNAi line targeting *wrd* and *wdb* was generated as follows. A 1107-bp fragment [nucleotides 4557–5663 of pNP (Qiao et al., 2018)] containing MCS2 and the linker between MCS1 and MCS2 was synthesised by IDT and inserted between the EcoRI and HpaI sites of pWalium20 by Gibson assembly (NEBuilder Hifi DNA Assembly, NEB). In the resulting plasmid, the region between StuI and NheI containing loxP, gypsy, 5×UAS and the hsp70 promoter was replaced by Gibson assembly with a 1010 bp fragment containing loxP, gypsy, 5×UAS and the DSCP promoter, amplified from pNP using primers, 5′-AATTGATCCACTAGAAGGCCTAA-3′ and 5′-ACTGGGAAAACATCCATGCTA-3′. This resulting plasmid (pGP11) was verified by Sanger sequencing to confirm that no unwanted mutations had been introduced during PCR or gene synthesis. Two plasmids, each expressing a single shRNA targeting *wrd* or *wdb*, were generated by inserting the following pairs of complementary oligonucleotides with cohesive ends between the EcoRI and NheI sites of pGP11 using T4 DNA ligase (NEB). For *wdb*: 5′-CTAGCAGT<u>GGATGTTGAGCTGCAG</u>CAA<u>TTTAGTTATATTCAAGCATA</u>AATT<u>GCTGCAGCTCAACATCCG</u>CG-3′ and 5′-AATTCGC<u>GGATGTTGAGCTGCAGCAATTTATGCTTG</u>AATATAACTA<u>AATTGCTGCAGCTCAACATCC</u>ACTG-3′. For *wrd*: 5′-CTAGCAGT<u>GCAGCATCAGAAACGACAACA</u>TAGTTATATTCAAG<u>CATATGTTGTCGTTTCTGATGCTGC</u>GCG-3′ and 5′-AATTCGCGCAG<u>CATCAGAAACGACAACA</u>TATGCTTGAATATAACTA<u>TGTTGTCGTTTCTGATGCTGC</u>ACTG-3′ (underlined nucleotides form base pairs complementary to the transcripts).

The region between EcoRI and NheI was PCR-amplified from the *wrd* single shRNA plasmid using PrimeSTAR (Takara) with flanking overlapping sequences for Gibson assembly. The PCR product was then inserted into the SpeI site of the *wdb* single shRNA plasmid by Gibson assembly. The absence of unwanted mutations was confirmed by Sanger sequencing. Finally, the resulting plasmid was microinjected into the fly line (BDSC8622; P{CaryP}attP2) using φC31 integrase-mediated transgenesis, carried out by BestGene Inc.

### *Drosophila melanogaster* genetic techniques

Standard *Drosophila* techniques (Ashburner et al., 2011) were followed. To test the localisation of an endogenous protein using an antibody, *w¹¹¹⁸* was used as the wild type. To test the localisation of an exogenous GFP-tagged protein in oocytes, virgin females of the *GAL4*-driver line *P[GAL4::VP16-nos.UTR]MVD1* (BDSC_4937) were crossed with males carrying a transgene of the GFP-tagged protein under the control of *UASp*. Transgenic lines with various GFP-tagged spindle-associated proteins were previously reported (Costa and Ohkura, 2019).

Transgenic RNAi lines (BDSC38900, BDSC38901) obtained from the Bloomington *Drosophila* Stock Center were used for single knockdown of the *wrd* and *wdb* genes, respectively. These lines were crossed with a fly line carrying the GAL4-diver *P[mat-α4-tubulin-GAL4-VP16]V37* and *UASp-GFP-Mei-38*. For double knockdown of the *wrd* and *wdb* genes, a transgenic RNAi line carrying sequences for two shRNAs under *UASp* was generated in this study and crossed with the same GAL4 driver. As a no-RNAi control, the same GAL4 driver was crossed with *w¹¹¹⁸*.

### Taxol treatment and immunostaining of *Drosophila* oocytes

Newly eclosed female adult progeny (<1 day old) were matured at 25°C for 3 days with males and standard cornmeal medium supplemented with dried yeast powder. Ovaries from 24 mature females were dissected in buffer containing 80 mM PIPES pH 6.8, 1 mM MgCl₂, 1 mM EGTA and 20 μM taxol (Merck T7402). After incubation in the same taxol-containing buffer for a further 15 min, the excess buffer was removed and then 30 ml of fresh methanol were added. These methanol-fixed ovaries were used for immunostaining. The ovaries were sonicated with a microprobe using VibraCell (VCX500; Sonics) for a 1-s burst, and mature oocytes without the chorion and the vitelline membrane were collected and kept in fresh methanol. Sonication was sometimes repeated to collect enough oocytes. To rehydrate, collected oocytes were incubated in a 400 μl of PBS/methanol (1:1) for 10 min, followed by incubation in 400 μl of 100% PBS for 10 min. The oocytes were incubated with 100 μl of the blocking buffer (PBS, 0.1%

Triton X-100 and 10% FBS) for 30 min. The oocytes were incubated for 4 h at 22°C or overnight at 4°C with primary antibodies diluted in 100 μl of the blocking buffer. After washing three times in 200 μl PBST (PBS containing 0.1% Triton X-100) for 10 min each, the oocytes were incubated for 2 h at 22°C with 100 μl of secondary antibodies plus 0.5 mg ml⁻¹ DAPI in PBST. This was followed by four 15-min washes in PBST and one in PBS. Oocytes were mounted in the mounting medium (85% glycerol and 2.5% propylgallate) between a coverslip and a glass slide sealed with nail varnish.

Antibodies were used for immunostaining as below: anti-α-tubulin (mouse monoclonal DM1A; 1:250; Sigma-Aldrich), anti-TACC (rabbit polyclonal against D-TACC-CTD; 1:1000; Loh et al., 2012), anti-Ncd (rabbit polyclonal against the full-length, 1:1000), anti-GFP (rabbit polyclonal; A11122; 1:125; Thermo Fisher Scientific), anti-Mink (rat polyclonal; 1:500; Syred et al., 2013) and anti-Aurora B (rat polyclonal; 1:1000; generated in this study). Secondary antibodies conjugated to Alexa Fluor 488, Cy3 and Cy5 were used (1:100–1:200; The Jackson Laboratory or Molecular Probes).

After immunostaining, *Drosophila* oocytes were imaged using an Axiovert Z1 microscope (Zeiss) attached to a confocal laser scanning head LSM800 (Zeiss) or an AxioObserver 7 microscope (Zeiss) attached to a LSM900 (Zeiss). Oocytes were visualised under a Plan-Apochromat objective lens (63×/1.4 numerical aperture) with Immersol 518F oil (Zeiss). Images were captured at 512×512 pixels (the pixel size 100 nm×100 nm) and at 0.5 μm Z-intervals. The AxioObserver 7 with LSM900 produced similar, but not quantitatively identical, results to those with the Axiovert Z1 with LSM800, including lower background signals. Only the images acquired by the same microscope were compared and presented in the same figure. Data presented in Figs 1, 2, Figs S1–S3, S5 and S6 were acquired with the AxiovertZ1 with LSM800; data presented in Figs 4–6, Fig. S8 were acquired with the AxioObserver7 with LSM900. The images are presented after maximum-intensity projection and a linear brightness enhancement.

### Image analysis and quantification

To quantify the distribution of GFP–Mei-38, the signal intensities of α-tubulin, GFP and DNA were measured in taxol-treated oocytes (Tables S1, S2). After a maximum-intensity projection, a line with 1 μm width was drawn along the long axis of each spindle and an ectopic microtubule cluster with a similar maximum tubulin intensity in the same oocytes, and the intensity (the mean of each 1-μm width) was measured in 0.01 μm intervals along the lines using ImageJ. For a line for ectopic microtubules, the position of the highest tubulin intensity was set to 0. For a spindle line, the position of the centre of the chromosomal mass was set to 0. When this position was different from the lowest tubulin signal, the mid-point between the two was set to 0. Then, each intensity value was normalised by dividing the maximal value on the line of the spindle in the same oocyte. The graphs show the mean intensity with the s.e.m. along the long axis of the spindle and the line across an ectopic microtubule cluster. When the profiles are compared, each profile of the Mei-38 mean signal intensity is presented after normalisation to make the mean background signal intensity (at 0 in the spindle line) identical to the control. For a two-tailed unpaired *t*-test, the highest Mei-38 signal intensity along each line drawn on the spindle or ectopic microtubules was compared to the control (at the top row of a figure).

### Immunoblotting of *Drosophila* ovaries

Twenty-four pairs of ovaries were dissected from mature females in absolute methanol. Methanol-immersed ovaries were washed with PBS once and then 200 μl of PBS were added. After ovaries were homogenised with a pestle, 200 μl of 2× SDS loading buffer (100 mM Tris-HCl pH 6.8, 4% SDS, 20% glycerol, 0.2% bromophenol blue) plus 5% β-mercaptoethanol were added. The mixture was boiled at 95°C for 3 min, and 10 μl was loaded on each lane of SDS PAGE gel (Mini Protean Tetra Biorad) for electrophoresis at 150 voltages for 1 h. Then, proteins were transferred from the gel onto nitrocellulose membranes (Protran 0.2 NC; GE Healthcare) under 100 voltages for 20 min. The membrane was stained with a reversible Protein Stain kit (Thermo Fisher Scientific). After destaining, it was incubated with PBSTw (PBS plus 0.1% Tween 20) containing 3% skim milk for 1 h. Then it was incubated with the primary antibodies in PBSTw containing 3% skim milk at 22°C overnight with rotation. After three 10-min washes in PBSTw, the membrane was incubated

with fluorescent secondary antibodies diluted in PBSTw at 22°C for 2 h with rotation. After four 15-min washes in PBSTw, the membrane was visualised on an Odyssey CLx imaging scanner (v3.0.30; LI-COR) at 600 ppi. The original uncropped images are shown in Fig. S9.

The primary antibodies for immunoblotting were anti-GFP (rabbit polyclonal; A11122; 1:1000; Thermo Fisher Scientific) and anti-α-tubulin (mouse monoclonal DM1A; Sigma-Aldrich; 1:2000). The secondary antibodies used were IRDye 800CW-conjugated goat anti-rabbit-IgG (1:20,000; LI-COR Biosciences, 926-32211), IRDye 800CW-conjugated goat anti-rat-IgG (1:20,000; LI-COR Biosciences, 926-32219) and IRDye 680LT-conjugated goat anti-mouse-IgG (1:15,000; LI-COR Biosciences, 926-68020).

### Live imaging of *Drosophila* Schneider S2 cells

S2 cells were cultured in Schneider medium (Invitrogen, 21720024) supplemented with 10% heat-inactivated fetal calf serum (Invitrogen, 10500064) and 1% antibiotics-antimycotics (Sigma, A5955) at 27°C. The cell line was a kind gift from Professor Patrick Heun's laboratory (Biology, TU Darmstadt, Germany). Although they were not recently authenticated or tested for contamination, this would not influence our interpretation of the results. To observe the localisation of GFP–Mei-38 and its variants, expression plasmids containing inserted genes were transfected using X-tremeGENE HP (Merck). After 24 h, images of live transfected cells were obtained using an Axiovert microscope (Zeiss) attached to a spinning-disc confocal head (CSU-X1; Yokogawa) controlled by Volocity software (PerkinElmer).

### Protein production and purification

To generate the antigen of Mei-38 or Aurora B antibody, we produced N-terminal maltose binding protein (MBP)-tagged Mei-38(1–278) or the full-length Aurora B. The entry plasmid pENTR-Mei-38(1-278) or -Aurora B was recombined using LR Clonase with a destination vector containing the Gateway cassette at the XmnI site of pMALc2.

*E. coli* [BL21(DE3)/pLysS] carrying the plasmid was cultured to saturation at 37°C and diluted 1:100 to 1000 ml of LB (Sambrook et al., 1989) with ampicillin and chloramphenicol (Sigma). After 2 h at 37°C, a final 1 mM IPTG (Sigma) was added to the culture. After further culturing at 37°C for 3 h, bacteria were centrifuged (17,000 $g$ for 10 min) and re-suspended in 5 ml lysis buffer (50 mM Tris-HCl pH 8, 1 mM EDTA, 100 mM NaCl) and lysed by repeated freeze-thaw cycles on dry ice and water baths until the cell suspension became viscous. After adding a final 250 µg ml$^{-1}$ lysozyme (Sigma), the lysate was incubated on ice for 10 min, then at 37°C for 10 min, followed by supplementing with final 10 µg ml$^{-1}$ DNase (Sigma) and 2 mM MgCl$_2$. It was left at 37°C for 10 min or longer until the viscosity decreased. 5 ml PBS was added to the lysate, which was then cleared by centrifugation at 14,000 rpm (17,000 $g$) using Eppendorf FA-45-18-11 for 10 min at 4°C. The supernatant was applied to a column containing 1 ml of amylose resin (NEB) prewashed with 5 ml of PBS, and the flowthrough was collected. After washing twice with 20 ml of PBS, the MBP-tagged proteins were then eluted through a column by adding a PBS containing 10 mM maltose. Five fractions of 500 µl were collected, and then 50 µl of each fraction was mixed with equal volume of 2× SDS loading buffer and final 5% β-mercaptoethanol. Samples were boiled for 3 min at 95°C and analysed by SDS-PAGE with 1 µg BSA for quantification. The first three eluted fractions from two rounds of purification were mixed together and then concentrated using the centrifugal filter (Merck Millipore) to get a final volume of 2 ml of 0.2 mg ml$^{-1}$. Two rats were immunised four times each by Eurogentec using 50 µg of the purified protein each.

To perform the *in vitro* microtubule-binding assay we produced Mei-38(176–242) with an MBP tag at the N-terminus. The entry plasmid pENTR-Mei-38(176-242) was recombined using LR Clonase with the same destination vector as MBP-Mei-38(1-278). MBP-Mei-38(176–242) was expressed and purified following the same procedure as MBP-Mei-38(1-278), except 500 ml of bacteria culture and 3 ml lysis buffer were used (Fig. S7).

### *In vitro* microtubule-binding assay

To test the microtubule-binding activity of MBP–Mei-38 (176–242 aa), 10 µl of 5 mg ml$^{-1}$ pure tubulin from porcine brains (T240: Cytoskeleton Inc) in BRB80 (80 mM PIPES pH 6.8, 1 mM MgCl$_2$, 1 mM EGTA)

containing 1 mM GTP was diluted with 40 µl of BRB80 buffer. 1.8 µg of freshly made MBP–Mei-38 (176–242) protein was mixed with diluted tubulin before addition of 10 µg BSA protein, and then BRB80 buffer up to a total volume 200 µl. After centrifugation at 22°C with 45,000 rpm using a Beckman TLA120.2 rotor for 20 min to remove insoluble proteins, the supernatant was split into two fractions of 50 µl each. To one fraction (sample), final 20 µM paclitaxel (Merck T7402) and 1 mM GTP were added, and the same volume of BRB80 was added to the other (control). Both samples were incubated at 22°C for 10 min and then centrifuged at 22°C at 14,000 rpm (17,000 $g$) using an Eppendorf FA-45-18-11 rotor for 15 min to pellet microtubules. The sample pellet was washed with 100 µl of BRB80 containing 20 µM paclitaxel and 1 mM GTP and the control pellet was washed with 100 µl of BRB80. After the pellets were resuspended in 25 µl of BRB80, 25 µl of 2× SDS sample buffer and final 5% β-mercaptoethanol were added to the supernatants and pellets. They were boiled and 10 µl were loaded on each lane for western blotting using an anti-MBP antibody (rat polyclonal against the full length; 1:1000) as above.

### Software usage

The software used in this study are commercially or publicly available: ZEN (Zeiss), Volocity (Quorum Technologies Inc), ImageJ (NIH), Excel (Microsoft), Prism 10 (GraphPad). Edinburgh (access to) Language Models (ELM) services were used to help correcting or rephrasing English on some occasions during writing of the manuscript. After using these services, the authors reviewed and edited the content as needed and take full responsibility for the content of the publication.

### Acknowledgements

We are grateful to the current and past members of the Ohkura laboratory for their help and support. We also thank Prof Jian-Quan Ni and Nobert Perrimon for kindly sharing the pNP plasmid. Stocks from the Bloomington Drosophila Stock Center (National Institutes of Health grants P40OD018537) were used in this study. A part of the work was carried out in the former Wellcome Centre for Cell Biology, which was supported by the Wellcome Trust [203149].

### Competing interests

The authors declare no competing or financial interests.

### Author contributions

Conceptualization: X.W., H.O.; Data curation: X.W., H.O.; Formal analysis: X.W., H.O.; Funding acquisition: H.O.; Investigation: X.W., H.O.; Methodology: X.W., C.F.C., H.O.; Resources: G.P., I.D., A.C.; Supervision: H.O.; Visualization: X.W., H.O.; Writing – original draft: X.W., H.O.; Writing – review & editing: X.W., G.P., H.O.

### Funding

This work was supported by the Wellcome Trust [206315, 227907 to H.O.]. X.W. received a PhD studentship from Darwin Trust of Edinburgh. Open Access funding provided by University of Edinburgh. Deposited in PMC for immediate release.

### Data and resource availability

All reagents generated in this study, including fly lines and plasmids, are available on request. All relevant data and details of resources can be found within the article and its supplementary information.

### First Person

This article has an associated First Person interview with the first author of the paper.

### Peer review history

The peer review history is available online at https://journals.biologists.com/jcs/lookup/doi/10.1242/jcs.264161.reviewer-comments.pdf

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
