## [Peer Review File · Journal of Cell Science]

Identification of locally activated spindle-associated proteins in oocytes uncovers a phosphatase-driven mechanism

Xiang Wan, Gera Pavlova, C. Fiona Cullen, Igor Dasuzhau, Aleksandra Ciszek and Hiroyuki Ohkura

DOI: 10.1242/jcs.264161

Editor: Renata Basto

Review timeline

Original submission:	23 May 2025
Editorial decision:	7 July 2025
First revision received:	17 September 2025
Accepted:	29 September 2025

Original submission

First decision letter

MS ID#: jcs.264161

MS TITLE: Identification of locally activated spindle-associated proteins in oocytes

AUTHORS: Hiroyuki Ohkura; Xiang Wan

ARTICLE TYPE: Research Article

Dear Hiro,

We have now reached a decision on the above manuscript.

To see the reviewers' reports and a copy of this decision letter, please go to:

As you will see, the reviewers¹ and 2, raise important points considering the function of Mei-38. We would love to invite you to provide some extra data considering the localization of Mei-38 in the phosphatase mutant/depletion background. Please also take into consideration the other suggestions to improve clarity. If you think that you can deal satisfactorily with the criticisms on revision, I would be pleased to see a revised manuscript.

Reviewer 1

Spatial regulation of spindle assembly factors (SAFs) is critical during meiotic spindle assembly in oocytes, as microtubules must be organized into a bipolar spindle only around the chromosomes, which occupy a very small region within the extremely large cytoplasm. This study developed an assay to assess whether each SAF localizes preferentially to spindle microtubules or uniformly to cytoplasmic microtubules in *Drosophila* oocytes. The authors identified nine SAFs that specifically localize to spindle microtubules, even when cytoplasmic microtubules are artificially stabilized and clustered by taxol treatment. They then focused on Mei-38, the *Drosophila* homolog of TPX2, one of the best-studied SAFs. They provide evidence that a specific site in Mei-38 (Ser-225) is key to localizing this protein to spindle microtubules and propose a role for PP2A-B56 in this process.

The authors address an important question in meiosis and present intriguing observations on the spatial regulation of SAFs. Some of the data are somewhat complex, and certain results differ

between full-length and truncated Mei-38. The authors carefully discuss the possibility of multifaceted regulation. The findings are of broad interest, as a similar mechanism may also operate in mammalian oocytes. I would like to see this study published in J Cell Sci. However, the molecular interpretation of the key finding is not yet fully supported by the current data. The authors need to provide stronger evidence to support their conclusion.

Major issue:

The evidence supporting PP2A-B56 involvement is not strong. The putative docking motif within Mei-38 appears dispensable for spindle localization of the full-length protein. Biochemical evidence that PP2A-B56 physically interacts with Mei-38 is lacking, as is direct analysis of the phosphorylation or dephosphorylation status of the S225 residue. Additional experimental support is needed to conclude that Mei-38 is regulated in a PP2A-B56-dependent manner. For example, is Mei-38 localization affected in a PP2A-B56 mutant background? This could be a relatively straightforward experiment, as RNAi lines targeting PP2A-B56 appear to be available.

Other issue:

Page 12, line 6 and page 13, line 10: Nine spindle-associated proteins?

Reviewer 2

SUMMARY OF THE ADVANCE MADE IN THIS PAPER AND ITS POTENTIAL SIGNIFICANCE TO THE FIELD

The manuscript by Wan and Ohkura reports new insights into how spindles assemble in oocytes, and specifically, into how spindle proteins are spatially regulated such that microtubule assembly is promoted in the vicinity of chromosomes. The authors use a clever approach to identify proteins that are spatially regulated in *Drosophila* oocytes, by adding taxol to stimulate microtubule polymerization away from chromosomes and then looking for proteins that are preferentially localized to spindle microtubules. They then go on to investigate one of these proteins, the TPX2 homolog Mei-38, in more detail, providing new insights into the regulation of this conserved protein. They identify a microtubule binding domain in this protein that is regulated spatially, and they identify a conserved serine and a PP2A-docking motif that is required for this regulation. Their data is consistent with a model in which PP2A dephosphorylates this serine, promoting microtubule binding of Mei-38, and thus spindle assembly, near chromosomes.

This paper is interesting and reports novel findings that should be of interest to the cell division community. In addition, the data is generally solid and convincing, and the work will stimulate future studies, both into the spatially-regulated proteins that are not examined in detail in this study, and also into Mei-38. I have a few minor suggestions that would improve this promising study prior to publication (and I also found some errors in my readthrough that should be corrected).

SUGGESTIONS TO AUTHORS

Specific points:

- Page 7 line 8: In the sentence listing the 9 proteins identified, there is a figure callout to Figure 1C, but I think this is incorrect. The data related to these proteins is in Fig 1D, S2, S3, and 2B - please fix the callout to reference the correct data.
- Figure 1: This is very minor, but I was not sure why the labels on the graphs in Figure 1C were different from the graphs in Figure 1E. I would probably change the labels on 1C to "spindle" and "ectopic" for consistency throughout the manuscript.
- The authors note that Mei-38(195-323) failed to localize to any microtubules, but then also explain that this is likely due to very low protein levels (page 8 lines 20-21). Because this data is inconclusive (it is unclear if the protein could localize to microtubules, if the protein level was higher), I think that it should be removed from Figure 2, both from the schematic in 2A, and the data in 2C. I think this would be better because, if a reader was to look at this figure without carefully reading the caveat presented in the results section, they might assume that this data was more solid. It is fine to keep the 195-323 data (currently in Figure 2C) in the manuscript, but I would put it together with the Western blot in Figure S6, so that it is not mis-interpreted.
- For the proteins that were reported but not examined in detail in this study (Kinesin-13 and Kinesin-5 in Figure S1, the CPC components in Figure S2, and Ncd and Cyclin B in Figure S3), it would be helpful for the authors to report the number of oocytes they examined to generate these

data. I recognize that these proteins are not the focus of this study (and so they do not need to be as rigorously quantified as the proteins in Figure 1C and 1E). However, it would be useful to report the N's in the legends for Figures S1-S3, so the reader has a better sense of whether these data are solid or preliminary. (If I missed this information somewhere, I apologize!).

- I recognize that the uncropped blots in Figure S8 were provided for full data transparency, as required by many journals, but it would be nice to briefly state in the figure legend what the lanes that were cropped out represent. Specific protein details are not essential, but something general like "lanes that were cropped out represent other protein fragments that were not included in our final analysis" or "lanes that were cropped out represent duplicate conditions...", etc., would be nice for clarity.

Typos/wording suggestions:

- Abstract line 10: suggest changing to "identified a microtubule-binding domain in this protein..."
- Page 3 line 10: "the spindle assembly" should be "spindle assembly"
- Page 3 line 24: should be "Kinesin-5"
- Page 4 line 6: should be "mutants affect spindle organization..."
- Page 6, line 24: should be "Kinesin-13 Klp10A"
- Page 7 line 27: There is a missing ")" after "(Fig. 2B"
- Page 8 line 5, and line 28: The Goshima reference is listed twice in each of these instances.
- Page 10: "the spindle localization" should be "spindle localization"
- Page 11, line 10: "consistent to" should be "consistent with"
- Page 14 line 19: should be "serves as a new..."

First revision

Author response to reviewers' comments

[Reviewer 1]

"The authors address an important question in meiosis... The findings are of broad interest, as a similar mechanism may also operate in mammalian oocytes.. I would like to see this study published in J Cell Sci." We are very pleased to hear very positive comments and constructive suggestions which we have fully incorporated to the revised manuscript.

Major issue

"...Additional experimental support is needed to conclude that Mei-38 is regulated in a PP2A-B56-dependent manner. For example, is Mei-38 localization affected in a PP2A-B56 mutant background?"

We have carried out the suggested experiment. Indeed, Mei-38 localisation to the spindle was significantly reduced when two isoforms of PP2A-B56 (*wrd*, *wdb*) were simultaneously knocked down. Although this double knockdown result supports our hypothesis, single knockdowns showed either no change (*wrd*) or an increase (*wdb*), suggesting a more complex picture. As these changes are relatively small, the RNAi efficiencies are unknown, and PP2A-B56 has many other substrates, we have included these results (Fig. S8; attached) in the revised manuscript without making a conclusive statement.

Other issue:

"Page 12, line 6 and page 13, line 10: Nine spindle-associated proteins?"

Thank you for pointing out our errors. They have been corrected.

[Reviewer 2]

"This paper is interesting and reports novel findings that should be of interest to the cell division community. In addition, the data is generally solid and convincing, and the work will stimulate future studies..."

We are very pleased to hear very positive comments and constructive suggestions which we have fully incorporated to the revised manuscript.

"Page 7 line 8: In the sentence listing the 9 proteins identified, there is a figure callout to Figure 1C, but I think this is incorrect. The data related to these proteins is in Fig 1D, S2, S3, and 2B - please fix the callout to reference the correct data."

Thank you for pointing out the errors. They have been corrected.

"Figure 1: This is very minor, but I was not sure why the labels on the graphs in Figure 1C were different from the graphs in Figure 1E. I would probably change the labels on 1C to "spindle" and "ectopic" for consistency throughout the manuscript."

We have changed the labels in Fig. 1C to make them consistent to other figures, as suggested.

"...Mei-38(195-323) I think that it should be removed from Figure 2, both from the schematic in 2A, and the data in 2C.. It is fine to keep the 195-323 data (currently in Figure 2C) in the manuscript, but I would put it together with the Western blot in Figure S6, so that it is not misinterpreted."

We agree. As suggested, we have moved Mei-38(195-323) from Fig. 2, S5 to Fig. S6.

"For the proteins that were reported but not examined in detail in this study... it would be useful to report the N's in the legends for Figures S1-S3..."

We have included the oocyte numbers in the legends, as suggested.

"...the uncropped blots in Figure S8... it would be nice to briefly state in the figure legend what the lanes that were cropped out represent."

We have included the information on cropped-out lanes (strains or samples we did not use) in the legends, as suggested.

"Typos/wording suggestions:

- Abstract line 10: suggest changing to "identified a microtubule-binding domain in this protein..."
- Page 3 line 10: "the spindle assembly" should be "spindle assembly"
- Page 3 line 24: should be "Kinesin-5"
- Page 4 line 6: should be "mutants affect spindle organization..."
- Page 6, line 24: should be "Kinesin-13 Klp10A"
- Page 7 line 27: There is a missing ")" after "(Fig. 2B"
- Page 8 line 5, and line 28: The Goshima reference is listed twice in each of these instances.
- Page 10: "the spindle localization" should be "spindle localization"
- Page 11, line 10: "consistent to" should be "consistent with"
- Page 14 line 19: should be "serves as a new..."

Thank you so much for these suggestions. We have incorporated all of them.

[Reviewer 3]

"Overall, this is a very comprehensive, multi-scale study. The tools used are sophisticated and carefully chosen. The presentation of the different results is particularly effective, and the quantification is done in a careful manner. This is a very nice study, perfectly suited for journal of Cell Sciences. It deserves to be published after minor revisions."

We are very pleased to hear very positive comments and constructive suggestions which we have fully incorporated to the revised manuscript.

Major point:

"...PP2A depletion can be induced using RNAi lines that are available. It would be interesting to test the effect of this on both the full-length version of Mei-38 and its truncated form (176-278). No matter what form it takes, this result will always be of interest."

We have carried out the suggested experiment. Indeed, Mei-38 localisation to the spindle was significantly reduced when two isoforms of PP2A-B56 (*wrd*, *wdb*) were simultaneously knocked down. Although this double knockdown result supports our hypothesis, single knockdowns

showed either no change (*wrd*) or an increase (*wdb*), suggesting a more complex picture. As these changes are relatively small, the RNAi efficiencies are unknown, and PP2A-B56 has many other substrates, we have included these results (Fig. S8; attached) in the revised manuscript without making a conclusive statement.

Unfortunately we could not test effects of PP2A-B56 RNAi on the truncated form (176-278) due to technical issues (a fly line carrying the 176-278 and the driver was not healthy).

Minor comments

"For the purposes of clarity, it would be helpful to present the list of the 12 candidates tested at the beginning of Figure 1."

As suggested, we have included the list of the 12 candidates tested in Fig. 1.

"In the results section on page 10, line 5, the mutation of motif B56-3A is shown in panel C of Figure 4. It should be mentioned in the text as Fig. 4A, C."

As suggested, we have included a reference to both Fig. 4A and C.

Second decision letter

MS ID#: jcs.264161R1

MS Title: Identification of locally activated spindle-associated proteins in oocytes uncovers a phosphatase-driven mechanism

Authors: Xiang Wan; Gera Pavlova; C. Fiona Cullen; Igor Dasuzhau; Aleksandra Ciszek; Hiroyuki Ohkura

Article Type: Research Article

Dear Hiro,

I am happy to tell you that your manuscript has been accepted for publication in Journal of Cell Science, pending standard publication integrity checks.